# Effect of Combined Methamphetamine and Oxycodone Use on the Synaptic Proteome in an *In Vitro* Model of Polysubstance Use

**DOI:** 10.3390/genes13101816

**Published:** 2022-10-08

**Authors:** Daniel Meyer, Pranavi Athota, Austin Gowen, Nghi M. Nguyen, Victoria L. Schaal, Sowmya V. Yelamanchili, Gurudutt Pendyala

**Affiliations:** 1Department of Anesthesiology, University of Nebraska Medical Center, Omaha, NE 68198, USA; 2Department of Genetics, Cell Biology, and Anatomy; University of Nebraska Medical Center, Omaha, NE 68198, USA; 3National Strategic Research Institute, University of Nebraska Medical Center, Omaha, NE 68198, USA; 4Child Health Research Institute, University of Nebraska Medical Center, Omaha, NE 68198, USA

**Keywords:** striatin-1, polysubstance use, speedballing, methamphetamine, oxycodone

## Abstract

Polysubstance use (PSU) generally involves the simultaneous use of an opioid along with a stimulant. In recent years, this problem has escalated into a nationwide epidemic. Understanding the mechanisms and effects underlying the interaction between these drugs is essential for the development of treatments for those suffering from addiction. Currently, the effect of PSU on synapses—critical points of contact between neurons—remains poorly understood. Using an in vitro model of primary neurons, we examined the combined effects of the psychostimulant methamphetamine (METH) and the prescription opioid oxycodone (oxy) on the synaptic proteome using quantitative mass-spectrometry-based proteomics. A further ClueGO analysis and Ingenuity Pathway Analysis (IPA) indicated the dysregulation of several molecular functions, biological processes, and pathways associated with neural plasticity and structural development. We identified one key synaptic protein, Striatin-1, which plays a vital role in many of these processes and functions, to be downregulated following METH+oxy treatment. This downregulation of Striatin-1 was further validated by Western blot. Overall, the present study indicates several damaging effects of the combined use of METH and oxy on neural function and warrants further detailed investigation into mechanisms contributing to synaptic dysfunction.

## 1. Introduction

Substance use disorder (SUD) is often a long-term chronic disease that significantly impacts the central nervous system (CNS), and it is characterized by cognitive and behavioral dysfunction [1]. While SUD as a whole has been an ongoing challenge for healthcare and research, an issue further compounding prognosis in clinical practice is the large number of individuals with SUD presenting a history of a complex pattern of polysubstance use (PSU). Most notable is the simultaneous and combined use of a stimulant (cf. cocaine and METH) along with a depressant (cf. alcohol and opioid), which is often fatal [2]. This practice of combining a stimulant and an opioid is known as “speedballing” and is reported to occur for various reasons. Often, individuals become inured to the effects of a drug after prolonged use and will try combining an additional substance to experience their initial level of drug-induced high. More specifically, one of the most prevalent motivators for speedballing is the desire to offset the deleterious effects of individual drugs with a psychoactive opposite. This attempt to curb negative side effects magnifies drug addiction tendencies and may result in neurological effects that are greater than the sum of each drug individually [3]. Indeed, from 1999 to 2020, the number of overdose deaths caused by the combination of psychostimulants, primarily METH, and opioids increased from 187 to 14,777 [2]. This alarming rise in the number of deaths highlights the significant impact that speedballing can have on the brain and the need for a better understanding of the effects of long-term PSU on the CNS.

The present study focuses on the impact of combinatorial exposure to methamphetamine (METH) and oxycodone (oxy) on primary neurons. METH, one of the most potent psychostimulants, has widespread effects across the CNS, such as rapidly increasing the levels of dopamine and the inducement of irreversible damage to synapses [4,5]. Users ingest METH in shorter intervals or increased concentrations to reach the same high as they build a tolerance to the effects [6]. The chronic use of METH has been known to precipitate violent tendencies, psychosis, and depression [7]. Additionally, chronic METH users may experience memory deficits and impaired cognitive function, prompting users to continue abusing it with the purpose of feeling more alert [4].

Oxycodone, a semi-synthetic opioid, is widely prescribed for varying types of pain and primarily binds to mu-opioid receptors [8]. In 2020, the opioid dispensing rate per 100 people was 43.3 [9]. This high opioid prescription rate indicates that oxy, one of the most popular prescription opioids, is more accessible than illegal substances such as METH. Notably, chronic oxy users are likely to experience brain fog, impaired memory, and dizziness [10]. Furthermore, opioids can act as gateway drugs, meaning that patients have an increased likelihood of abusing and becoming addicted to oxy and other drugs after they have recovered from their physical injury [10]. Similar to METH, oxy abuse has been associated with causing synaptic damage [11].

Both drugs have been shown to bring about various adverse effects on health when misused separately, but, currently, there is little research examining the effects of METH and oxy when used in conjunction. Moreover, a current knowledge gap exists regarding how METH and oxy in combination can induce alterations at the synapse, including the lack of reliable markers. The application of high-throughput “omics” approaches has helped researchers to understand the role of synaptic proteins in disease states, such as drug addiction, as a proteomic analysis allows for the global view of drug-induced changes within a specific proteome [12,13]. Using quantitative mass-spectrometry-based proteomics on primary neurons exposed to METH and oxy, we identified a key synaptic protein, Striatin-1, to be downregulated. We post-validated this finding and further affirmed the role of Striatin-1 as a key player in the pathophysiological effects of speedballing on neurons via a pathway analysis. Furthermore, the identification of critical molecular and functional pathways predominantly associated with neural plasticity point to the damaging effects of speedballing at the synapse. In summary, the findings from this study not only provide insights into the effects of PSU in the brain but also identify potential targets for developing therapeutic strategies to mitigate the adverse outcomes associated with PSUs.

## 2. Materials and Methods

### 2.1. Animals and Ethics Statement

All pregnant Sprague Dawley rats used in the experiment were purchased from Charles River Laboratories Inc. (Wilmington, MA, USA), and they were housed under a 12 h light–dark cycle and fed ad libitum. All procedures and protocols were conducted in accordance with the National Institutes of Health Guide for the Care and Use of Laboratory Animals and approved by the Institutional Animal Care and Use Committee of the University of Nebraska Medical Center (IACUC protocol: 17-080 approved 1 June 2021).

### 2.2. Primary Rat Neuronal Cultures

Cortical neurons from embryonic day 18 (E18) Sprague Dawley rats were plated at a density of 8.5 × 10^5^ cells per well onto 6-well plates previously coated with poly-d-lysine. The cells were cultured in Neurobasal media containing 0.5 mM L-glutamine and B27 supplement (Life Technologies, Carlsbad, CA, USA) at 37 °C in a humidified atmosphere of a 5% CO_2_ incubator. Every three days, a half exchange was performed on the medium. After 14 days in vitro (DIV 14), the cells were treated with 100 mM METH and 100 mM oxy for 24 h, following which the cells were collected in ice-cold PBS, and proteins were isolated using an NPer kit (ThermoFisher, Waltham, MA, USA).

### 2.3. Mass Spectrometry Analysis and Protein Identification

Protein quantification was performed using a Pierce BCA protein assay (Thermo Scientific, Rockford, IL, USA), and the procedure described in our earlier studies was followed [4,14,15,16,17,18]. The mass spectrometry analysis was conducted using a UNMC Mass Spectrometry Core (Omaha, NE, USA), and the protocol was based on the label-free quantitative mass spectrometry protocol described in our recently published studies with minor modifications [11,14,15]. Specifically, we used 50 µg of protein per sample (n = 3/group).

Protein identification also followed the procedure established in our lab’s previous studies [11,14,15]. Briefly, the in-house mascot 2.6.2. (Matrix Science, Boston, MA, USA) search engine was used to further explore the proteins from tandem mass spectrometry data. The search targeted full tryptic peptides and allowed for two missed cleavage sites. The carbamidomethylation of cysteine was set as a fixed modification. The acetylation of protein N-terminus and oxidized methionine were included as variable modifications. The highest allowed fragment mass error was 0.02 Da, and the precursor mass tolerance threshold was set at 10 ppm. A false discovery rate (FDR) of ≤1% was used to calculate the significance threshold of the ion score. A qualitative analysis was performed using progenesis QI proteomics 4.1 (Nonlinear Dynamics, Milford, MA, USA). 

### 2.4. Bioinformatics Analysis

Proteins were recognized as differentially expressed if the *p*-value of the *t*-test was ≤0.05 and the absolute fold change was ≥1.5. A gene ontology (GO) analysis of differentially expressed proteins (DEPs) was conducted using the Cytoscape plug-in ClueGO [16]. Biological processes and molecular functions were included for the GO enrichment analysis. A canonical pathway analysis was performed using Ingenuity Pathway Analysis (IPA) software (Ingenuity^®^ Systems, Redwood City, CA, USA, www.ingenuity.com, accessed on 21 June 2022) by comparing the DEPs against known canonical pathways (signaling and metabolic) within the IPA database. For further analysis, enriched pathways with a Benjamini–Hochberg false discovery rate (FDR) *p*-value ≤ 0.05 were considered.

### 2.5. Western Blot

Western blotting was conducted as described in our previous studies [11,14,17,18,19]. Briefly, cell lysates from both the control and METH+oxy treatment were loaded onto 4–12% Bis-Tris wells (Invitrogen, Waltham, MA, USA) under reducing conditions. The gels were then transferred to a nitrocellulose membrane using iBlot2 (Invitrogen, Waltham, MA, USA) and immunodetection. Nonspecific antibody blocking was carried out using Superblock (ThermoFisher, Waltham, MA, USA). Immunoblotting was performed with primary antibodies overnight at 4 °C against STRN1 (1:1000, ProteinTech, Rosemont, IL, USA. Cat No.: 21624-1-AP) and GAPDH (1:2500, Invitrogen, Waltham, MA, USA. Cat No.: MA5-15738). This was followed by secondary antibodies (1:2500, HRP-conjugated anti-rabbit IgG; Thermo Scientific, Waltham, MA, USA. Cat No.: G21234 and 1:5000, HRP-conjugated anti-mouse IgG; Sigma-Aldrich, St. Louis, MO, USA. Cat No.: A9044). Primary and secondary antibody dilutions were performed according to the manufacturer’s suggestions. Blots were developed on the Azure CSeries Imager (Azure Biosystems, Dublin, CA, USA) with SuperSignal West Pico Chemiluminescent Substrate (Thermo Scientific, Waltham, MA, USA). Band intensities were measured by ImageJ using the Gel Analysis function with the removal of background at the base of the peak intensity. After identifying the band intensities on each antibody, an individual band from Striatin was normalized in correspondence to the respective GAPDH. The average normalization value for each treatment (control vs. experimental) was calculated, and the fold change was determined by the ratio between the control and experimental groups.

### 2.6. Statistical Analysis

For the proteomic analysis, an unpaired *t*-test was performed post-normalization to detect significantly differentially expressed proteins following METH+oxy exposure. Proteins with ≥1 unique peptide and *p*-value < 0.05 were considered significant. Following Western blots and normalization by GAPDH, outliers were discovered and removed utilizing the ROUT method, with a maximum desired false discovery rate of 1%. The downregulation of striatin was determined to be statistically significant via an unpaired *t*-test following Welch’s correction, with *p* < 0.05. A statistical analysis was conducted using GraphPad Prism version 8.3.0 (LA Jolla, CA, USA).

## 3. Results

To elucidate the effects of METH+oxy on the proteome, we analyzed DIV14 neurons via high-throughput quantitative MS-based proteomics, which identified a total of 3125 proteins. The initial criterion of analysis included >1 unique peptide. However, in order to expand the potential networks of connection, the criterion of analysis was adjusted to ≥1 unique peptide and *p* < 0.05. We identified 94 DEPs. Of these, 54 were downregulated, and 25 were upregulated (Figure 1 and Appendix A).

We then utilized the ClueGO analysis to identify which key biological processes and molecular functions were associated with these DEPs. Of the upregulated functions and processes, the most abundant was S-nitrosoglutathione binding, which accounted for 37.5% of the gene ontology (GO) terms. Furthermore, the remainder of the GO terms, such as pyruvate kinase complex, early phagosome membrane, and MIT domain binding, equally accounted for 12.5% of the GO terms (Figure 2A). For the downregulated functions and processes, the one associated with the highest number of GO terms was the regulation of ferrochelatase activity, which accounted for 23.08%. The next most abundant terms were the regulation of phospholipase D activity and GMP reductase activity, each accounting for 11.54%. The postsynaptic intermediate filament cytoskeleton and cytosolic dipeptidase activity accounted for 7.69% each. Sulfur dioxygenase activity, fumarate hydratase activity, and various others encompassed 3.85% of the terms each (Figure 2B).

Utilizing the Ingenuity Pathway Analysis (IPA), we then analyzed potentially affected pathways involving the DEPs. Certain pathways, such as NRF2-mediated oxidative stress response, IL-8 signaling, and xenobiotic metabolism signaling, were activated following METH+oxy treatment. Others, such as the synaptogenesis signaling and Rho Family GTPase signaling pathways, were deactivated (Figure 3). Overall, these results indicate a disruption of neural function via the dysregulation of key pathways, biological processes, and molecular functions.

Due to the large number of possible hits generated by the high-throughput proteomics, further validation was necessary. Our analysis revealed a trend of the dysregulation of processes, functions, and pathways involved in neural plasticity and structural development. Moreover, ClueGO demonstrated the downregulation of proteins involved in the postsynaptic intermediate filament cytoskeleton, and IPA revealed the downregulation of the synaptogenesis signaling pathway. Therefore, we focused on validating the hits involved in these functions and processes. Further investigation of IPA revealed that a large number of DEPs within the synaptogenesis signaling pathway were explicitly associated with neuritogenesis. Striatin-1, a protein identified in our proteomics screen, was downregulated −7.7-fold post-METH+oxy treatment. Its expression was further validated in the cell lysates using Western blot (Figure 4).

## 4. Discussion

SUD is characterized by sensitivity to drug-associated cues and the motivation to maintain drug consumption, and it has been shown to be a chronic and relapsing disease [1]. With ~35 million people globally and 19 million people in the United States diagnosed with SUD, it is a significant public health issue, generating both social and economic costs. While drug use is more often studied in isolation, several of these drug users often simultaneously use multiple substances. This not only adds a layer of complexity to the understanding of how PSU impacts pathological outcomes but also hinders treatment options, increases drug relapse frequency, and raises mortality rates compared to the abuse of a single substance [20,21,22]. Users with long-term dependency on psychostimulants, such as cocaine and METH, have often been reported to exhibit a history of PSU [23]. Both cocaine and amphetamine users have been reported to co-use heroin, cannabis, tobacco, and alcohol [24,25]. Within the current ongoing opioid epidemic, emerging lines of evidence point to an escalated co-use of these two popular drug categories. Individuals diagnosed with SUD have shown increased aggression and irritability due to synaptic plasticity and impaired neurotransmitters caused by the chronic use of a single drug [26]. In the brain, the ventral tegmental area (VTA), the nucleus accumbens (NAc), and the medial prefrontal cortex (mPFC) are the regions that play an important role in reward processing and emotion regulation [26,27,28]. Psychostimulants can produce alterations in neuronal morphology and long-term disruptions to glutamate homeostasis, either over-stimulating or weakening glutamatergic inputs to the VTA region [29,30,31]. Similar to psychostimulants, opiates can strengthen glutamatergic inputs to the VTA dopaminergic neurons [32,33]. However, unlike psychostimulants, opioids decrease dendritic branching and spine density in the NAc regions [31,34,35]. Notably, electrical or chemical (i.e., glutamate analog) stimulation can also induce defensive rage behavior in mammals [36,37]. In addition, both opioids and psychostimulants have the ability to increase the dopamine level in the NAc [30,38]. Importantly, in animals, an increase in dopamine release in the NAc is associated with increased aggression [39]. Moreover, substance use can also influence changes in synaptic morphology, including dendritic spine branching and length [40,41], which all contribute to alterations in synaptic activity. Hence, the ability to disrupt synaptic activity could further alter neural circuitry and subsequently exacerbate violent behavior. While PSU further aggravates violent behavioral outcomes, the changes at synapses, including the lack of reliable synaptic markers contributing to such adverse outcomes, have not been completely understood.

The current study reveals important insights into the impact of PSU through changes in the synaptic proteome in an in vitro model of primary neurons. Our present study illustrated that the combined use of METH and oxy can cause DEPs involved in key neuronal pathways. Using the ClueGO analysis, we found associations between DEPs and various molecular functions and biological operations, including the regulation of ferrochelatase activity, phospholipase D activity, and S-nitrosoglutathione binding. Moreover, the IPA revealed significantly activated or deactivated pathways, such as cellular assembly and maintenance, as well as nervous system functional development that is associated with the DEPs in our METH+oxy group. Overall, these results provide foundational insight into the impact of acute METH and oxy exposure on neural function.

Our pathway analysis indicated the dysregulation of genes involved in cellular assembly, organization, function, and maintenance, specifically regarding cell adhesion, microtubule dynamics, and cytoplasmic organization (Appendix A). These characteristics have been shown in previous investigations to play a large role in both the acquisition of addiction, as well as the impairments caused by drugs of abuse. One such study demonstrated a loss of stable microtubules in striatal dopaminergic neurons after METH exposure, interfering with axonal transport and possibly leading to reversible deficits in dopaminergic markers, such as tyrosine hydroxylase, dopamine transporter, dopamine, and 3,4-dihydroxyphenylacetic acid (DOPAC) [42].

Furthermore, IPA revealed the dysregulation of 13 genes involved in neuritogenesis. (Appendix A). Addiction can be thought of as a kind of drug-induced neural plasticity, in which the brain gradually learns to rely on a chemical, leading to behavioral abnormalities and an impaired ability to regulate self-intake [28]. There are multiple mechanisms through which this process occurs. One such process is the drug-induced dysregulation of neuritogenesis [43]. Neuritogenesis is a process in which a specific arrangement of F-actin assembly creates a protrusion from the developing cell body, which then engages with microtubules and other components. Finally, protein consolidation allows for the development of neurites, such as axons and dendrites [44]. This process is crucial for the establishment of a mature nervous system capable of complex communication [45]. In the damaged hippocampus CA1 region, the promotion of neuritogenesis has been shown to rescue impairments in long- and short-term memory, cognition, and spatial intelligence [46]. Both METH and oxy have previously been shown to impair neural function. A study conducted by Martin et al. demonstrated that rats exposed to oxycodone exhibited impaired spatial intelligence during Barnes maze testing: traveling more slowly, crossing less distance, and less frequently finding the escape hole [47]. Moreover, exposure to oxycodone has been shown to diminish behavioral flexibility and cognitive ability [48]. Similarly, previous studies have shown that METH exposure can severely damage the neurons of the hippocampus, notably in the CA1 region, contributing to the loss of memory and spatial learning ability [49]. More generally, METH addiction has been linked to alterations in hippocampal morphology, deficits in learning and memory related to the hippocampus, and smaller limbic-related structures [50]. The results of our study present the dysregulation of neuritogenesis as a possible explanation for the mechanisms in which METH and oxy cause these impairments.

Along these lines, our proteomics screen identified the synaptic protein Striatin-1 to be downregulated following METH+oxy exposure. Previous studies have demonstrated that hippocampal neurons, which have been shown to play a prominent role in addiction, depend on Striatin-1 for neuritogenesis [51]. Striatin-1 is an intracellular caveolin and calmodulin-binding scaffolding protein belonging to the WD-repeat family [52]. An earlier study employing the anterograde tract-tracing labeling method demonstrated the subcellular localization of Striatin-1 to the dendritic spines containing the excitatory synapses in the rat striatum [53]. The downregulation of Striatin-1, as observed in our current study, potentially points to alterations in the dynamics of excitatory neurotransmission also associated with drugs of abuse. Another study demonstrated that Striatin-1 binds to caveolin-1, a scaffolding protein critical for regulating signaling and membrane trafficking [52]. Furthermore, another study demonstrated that striatin, when co-localized with phoecin, is implicated in vesicular trafficking machinery, specifically clathrin and dynamin-dependent membrane dynamics, in cerebellar and hippocampal synapses [54]. Based on our findings, we speculate that decreased Striatin-1 levels induced by the co-use of METH and oxy could indicate alterations in synaptic signaling, including membrane trafficking at the synapse. Further investigation into Striatin-1 and its associated interacting partners could further help in understanding its role in regulating synaptic function during PSU.

In summary, the present study elucidates that the co-use of METH and oxy impacts key molecular functions and biological processes within the CNS. The validation of the synaptic protein Striatin-1, which is associated with regulating essential neurological functions and processes, especially neural plasticity and synaptic structure, further contributes to its role as a potential target for therapeutic treatment to rescue defects inflicted by PSU.

## Figures and Tables

**Figure 1 genes-13-01816-f001:**
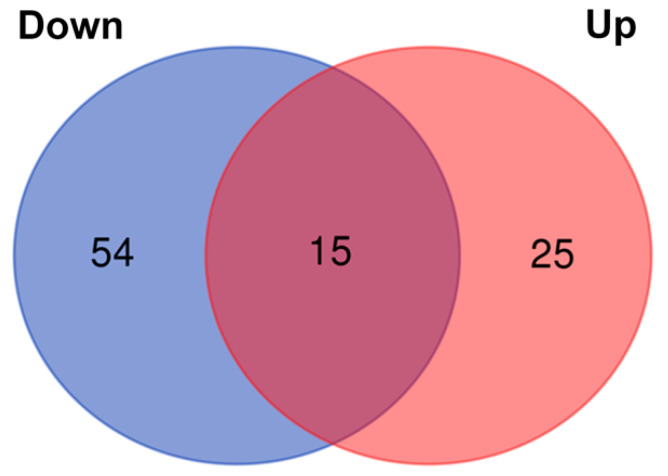
Venn diagram representing the differentially expressed proteins between control and METH+oxy groups.

**Figure 2 genes-13-01816-f002:**
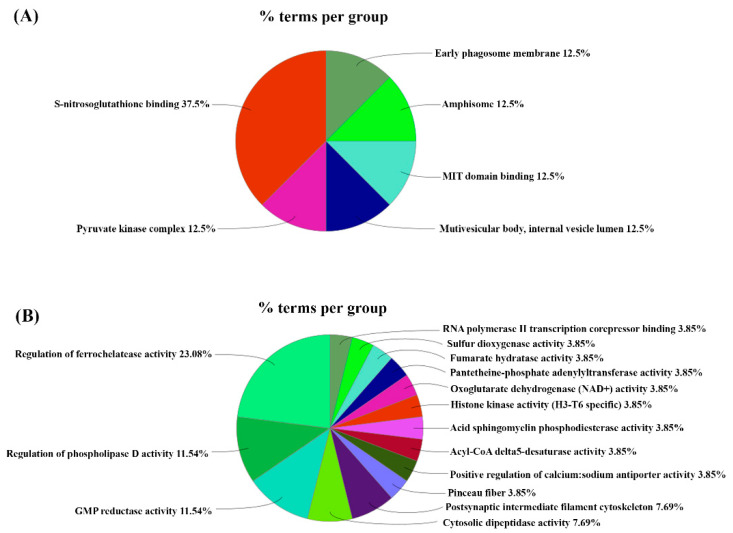
ClueGO analysis. Mapping of (**A**) upregulated and (**B**) downregulated molecular functions and biological processes.

**Figure 3 genes-13-01816-f003:**
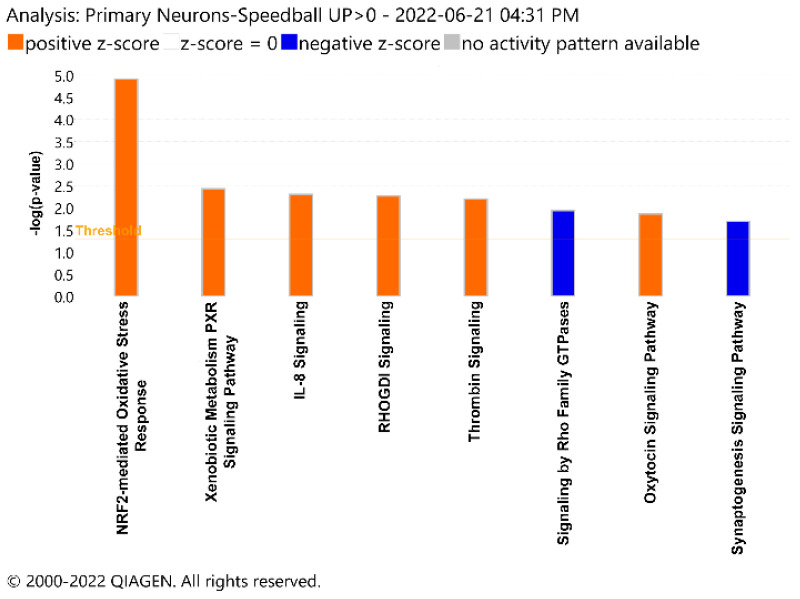
Ingenuity Pathway Analysis. Mapping of dysregulated pathways associated with METH/oxy treatment. Pathways are organized based on the negative log of the *p*-value. Positive and negative z-scores, represented by orange and blue coloration, suggest increased and decreased pathway activity, respectively.

**Figure 4 genes-13-01816-f004:**
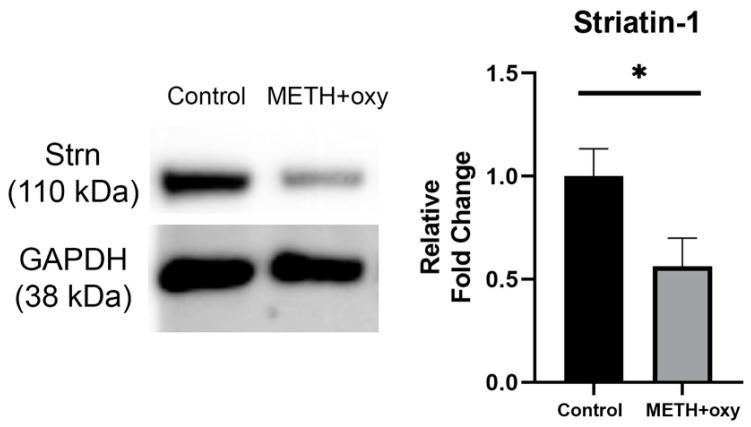
Validation of Striatin-1 downregulation. The downregulation of Striatin-1 was found to be statistically significant utilizing an unpaired *t*-test following Welch’s correction (* *p* < 0.05). Representative Western blots are displayed here. Relative fold-change data are depicted as mean ± SEM (n = 10).

## Data Availability

Data are contained within the article or Appendix A.

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
