# Peer review of "Effect of Combined Methamphetamine and Oxycodone Use on the Synaptic Proteome in an In Vitro Model of Polysubstance Use"

_genes, 2022, doi:10.3390/genes13101816_

Round 1
Reviewer 1 Report
Dear authors,
I am very satisfied about the outcome of this Ms, and I have only few comments.
Method:
Kindly, if they can provide the quantification method of the western blot bands!
Discussion:
Line 205, SUD characterized by ---, Just put is after SUD.
Paragraph 4, if you could write more previous studies that investigated the toxic effects of methamphetamine and oxycodone on neural function.
Lines 232 and 240, the authors mentioned (Table S3), but there is no table provided!
Author Response
Dear authors,
I am very satisfied about the outcome of this Ms, and I have only few comments.
Method:
Kindly, if they can provide the quantification method of the western blot bands!
Discussion:
Line 205, SUD characterized by ---, Just put is after SUD.
Paragraph 4, if you could write more previous studies that investigated the toxic effects of methamphetamine and oxycodone on neural function.
Lines 232 and 240, the authors mentioned (Table S3), but there is no table provided!
Dear reviewer,
Thank you for all of your comments and input in reviewing our manuscript. We have included the quantification and normalization methods in our methodology for the western blot (lines 152-159). Regarding line 205, we have addressed the wording and grammar accordingly.
Additionally, we elaborated further on the studies investigating the toxic effects of METH and oxy on neural function and cited those two sources (lines 293-294 and lines 303-305). Our apologies for the oversight in missing the supplemental Table 3 which we now have included in this revised version.
Reviewer 2 Report
The manuscript describes the effect of concurrent methamphetamine and oxycodone use on the synaptic proteome. Meyer et al. studied polysubstance use (PSU), an escalating national problem that has a deleterious impact on health and society. They utilized a suitable in vitro model and high throughput screening for molecular protein targets. Some points must be addressed before publication:
· Please add Cat. numbers for antibodies and kits that have been used.
· Supplementary blots showed some variability in loading based on the GAPDH blots; based on that, describe how band density was measured and normalized.
· The authors stated in lines 217 - 219, “While PSU further aggravates violent behavioral outcomes, changes at the synapses, including the lack of reliable synaptic markers contributing to such adverse outcomes, have not been completely understood.” Could you further explain the link between synaptic changes and PSU violent behavior?
Author Response
The manuscript describes the effect of concurrent methamphetamine and oxycodone use on the synaptic proteome. Meyer et al. studied polysubstance use (PSU), an escalating national problem that has a deleterious impact on health and society. They utilized a suitable in vitro model and high throughput screening for molecular protein targets. Some points must be addressed before publication:
Please add Cat. numbers for antibodies and kits that have been used.
Supplementary blots showed some variability in loading based on the GAPDH blots; based on that, describe how band density was measured and normalized.
The authors stated in lines 217 - 219, “While PSU further aggravates violent behavioral outcomes, changes at the synapses, including the lack of reliable synaptic markers contributing to such adverse outcomes, have not been completely understood.” Could you further explain the link between synaptic changes and PSU violent behavior?
Dear reviewer,
Thank you for all of your comments and input in reviewing our manuscript! We have added the catalog number for antibodies and kits used in the method section as suggested. For the western blot, we also included our band measurement and normalization methods in the western blot section under lines 152-159. Regarding the relationship between synaptic changes and PSU violent behavior, we have included a description (lines 239-255) on how psychostimulants and opioids could induce changes in neuronal morphology as well as long-term disruption in glutamate homeostasis that ultimately impact behavioral outcomes in such individuals.